# Fine Particulate Matter and Gaseous Compounds in Kitchens and Outdoor Air of Different Dwellings

**DOI:** 10.3390/ijerph17145256

**Published:** 2020-07-21

**Authors:** Célia Alves, Ana Vicente, Ana Rita Oliveira, Carla Candeias, Estela Vicente, Teresa Nunes, Mário Cerqueira, Margarita Evtyugina, Fernando Rocha, Susana Marta Almeida

**Affiliations:** 1Centre for Environmental and Marine Studies (CESAM), Department of Environment, University of Aveiro, 3810-193 Aveiro, Portugal; anavicente@ua.pt (A.V.); ritaanaoliveira@ua.pt (A.R.O.); estelaavicente@gmail.com (E.V.); tnunes@ua.pt (T.N.); cerqueira@ua.pt (M.C.); margarita@ua.pt (M.E.); 2Geobiosciences, Geotechnologies and Geoengineering Research Centre (GeoBioTec), Department of Geosciences, University of Aveiro, 3810-193 Aveiro, Portugal; tavares.rocha@ua.pt; 3Centre for Nuclear Sciences and Technologies (C2TN), Instituto Superior Técnico, University of Lisbon, Estrada Nacional 10, 2695-066 Bobadela, Portugal; smarta@ctn.tecnico.ulisboa.pt

**Keywords:** dwellings, indoor/outdoor, VOCs, carbonyls, PM_2.5_, OC/EC, morphology, PAHs

## Abstract

Passive diffusion tubes for volatile organic compounds (VOCs) and carbonyls and low volume particulate matter (PM_2.5_) samplers were used simultaneously in kitchens and outdoor air of four dwellings. PM_2.5_ filters were analysed for their carbonaceous content (organic and elemental carbon, OC and EC) by a thermo-optical technique and for polycyclic aromatic hydrocarbon (PAHs) and plasticisers by GC-MS. The morphology and chemical composition of selected PM_2.5_ samples were characterised by SEM-EDS. The mean indoor PM_2.5_ concentrations ranged from 14 µg m^−3^ to 30 µg m^−3^, while the outdoor levels varied from 18 µg m^−3^ to 30 µg m^−3^. Total carbon represented up to 40% of the PM_2.5_ mass. In general, the indoor OC/EC ratios were higher than the outdoor values. Indoor-to-outdoor ratios higher than 1 were observed for VOCs, carbonyls and plasticisers. PAH levels were much higher in the outdoor air. The particulate material was mainly composed of soot aggregates, fly ashes and mineral particles. The hazard quotients associated with VOC inhalation suggested a low probability of non-cancer effects, while the cancer risk was found to be low, but not negligible. Residential exposure to PAHs was dominated by benzo[a]pyrene and has shown to pose an insignificant cancer risk.

## 1. Introduction

We spend most of our time in indoor environments. As building infrastructures are increasingly airtight to save energy, epidemiological studies need to understand the extent to which outdoor levels of air pollutants persist as a determining factor of exposure and, consequently, of health in the indoor environment. Many of the outdoor pollutants are also prevalent within homes, contributing to unhealthy air. Kitchens are spaces not only for preparing meals, but also for socialising with family and friends, and where children often do their homework, and most of us watch television. The kitchen is indeed the heart of the home. Places where meals are made and eaten are considered microenvironments with specific characteristics [1]. After the bedroom, this indoor area is probably the room people spend the most time in. Air quality in a kitchen is influenced by many factors, such as the method of meal preparation and ingredients used, the cooking style, the temperature of the cooking process, the volume of the room, the efficiency of the exhaust hood, and the number of persons using the space [1,2,3]. However, in developing countries wood stove emissions are the main cause of kitchen-related air pollution in many deprived homes [4,5,6]. Nearly 3 billion people use solid fuels, especially biomass and coal, for cooking and heating and this number will continue to rise in the next decade [5] (and references therein). Since biomass-burning cookstoves are a noteworthy source of carbonaceous aerosols and gaseous compounds, this source of pollution has received increasing attention because emissions greatly contribute to the global burden of disease [5,7,8,9,10,11,12,13]. Estimates by the World Health Organisation (WHO) indicate that exposure to air pollution from cooking with solid fuels contribute to more than four million annual premature deaths globally, half a million of which are children under the age of 5 who die of pneumonia [14]. Emissions from biomass-burning cookstoves encompass products of incomplete combustion, such as volatile organic compounds (VOCs) and particulate matter with an aerodynamic diameter less than or equal to 2.5 μm (PM_2.5_), which can penetrate deeply into the alveolar sacs, where they can deposit and be absorbed, contributing to the entry of toxic substances into the bloodstream, including carcinogenic polycyclic aromatic hydrocarbons (PAHs) [15]. 

In the last two years, a substantial number of research articles has been published with the objective of documenting the indoor air quality in kitchens with low-efficiency biomass cookers, especially in underdeveloped countries. Some of these articles aimed at comparing emissions from traditional biomass stoves for household cooking with those from improved cookstoves [16,17,18,19,20,21,22,23,24,25,26,27,28,29,30,31,32,33]. In developed countries, in most cases, gas or electric stoves/cookers are used for meal preparation. Furthermore, a ventilation system mounted directly over the cooker/stove is an essential element in every kitchen to reduce the transport of odours and pollutants to neighbouring rooms. However, despite the pollutant levels in well-equipped modern kitchens are reportedly much lower, studies on this type of microenvironment are scarce and mostly focused on gaseous contaminants [34,35,36,37,38]. The WHO concluded that there is no convincing evidence of a difference in the hazardous nature of particulate matter from indoor sources as compared with those from outdoors and that the indoor levels are usually higher than the outdoor levels [39]. Continuous pressure to re-evaluate air quality standards stems from studies that have observed effects at low levels of particulate matter. These studies have suggested that, instead of mass concentration, some chemical components (e.g. carbonaceous compounds) may be a better metric for estimating the health risks [40]. A better understanding of indoor air pollutants, their levels and sources in specific microenvironments can help in adopting more efficient management strategies and mitigation measures to reduce health risks from exposure to PM_2.5_ and associated toxic constituents.

This study is based on a multi-pollutant monitoring campaign carried out in four biomass-free kitchens, for which studies are comparatively much scarcer, in order to answer the following questions: Are there significant differences in pollutant levels between modern kitchens equipped with gas ranges or electric hobs? Do the observed levels and compounds depend on housing factors or outdoor air? Are the risks resulting from inhalation of pollutants (VOCs and PM_2.5_-bound PAHs) routinely considered by international agencies of concern to health? Are these metrics sufficient to infer sources and effects or can the particle morphological analysis give us additional indications? The aim of this pilot study is not only to characterise air quality in a poorly studied microenvironment, such as kitchens, but also to draw lessons for conducting wider researches in the future.

## 2. Methodologies

### 2.1. Sampling and Analysis

A monitoring programme involving four kitchens with different characteristics (Table 1) and the respective outdoor air was conducted in the region of Aveiro, Portugal, in October and November 2017. Along with the neighbouring city of Ílhavo, Aveiro is part of an urban agglomeration that includes 120,000 inhabitants. Aveiro is located on the Atlantic coast, in the Central Region, at about 250 km to the North of Lisbon and 70 km to the South of Oporto. It is surrounded by beaches and by an extensive coastal lagoon. 

Low volume samplers were used to collect particulate matter (PM_2.5_) onto 47 mm diameter quartz filters. ECHO PM samplers (TCR Tecora, Cogliate, Italy) operating at 38.3 L min^−1^ were deployed in two residences, in which ten pairs of PM_2.5_ samples were collected for periods of 48 h. MiniVol^TM^ TAS samplers (Airmetrics, Springfield, OR, USA) were used in the other two dwellings, in which seven pairs of PM_2.5_ samples were collected for periods of 72 h at a flow of 5 L min^−1^. In the kitchens, the samplers were positioned near the dining tables in a central location. Outside, the equipment was placed on the porch, terrace or balcony adjacent to the kitchens. VOCs and carbonyls were sampled in parallel, also indoors and outdoors, using Radiello^®^ (Merck, Darmstadt, Germany) diffusive passive tubes (cartridges codes 145 and 165, respectively) in triplicate. Two consecutive samplings, each lasting 10 days, were performed at each site. VOCs were analysed by thermal desorption coupled to gas chromatography-mass at the Istituti Clinici Scientifici Maugeri (Pavia, Italy). The carbonyl-DNPH derivatives were analysed at the University of Aveiro by eluting with 2 mL of acetonitrile poured directly into the cartridge and stirring from time to time for 30 min. The extracts were filtered and then analysed in a high-performance liquid chromatography system (Jasco, Cremella, Italy) equipped with a PU- 980 pump, also from Jasco (Cremella, Italy), a manual injection valve (20 µL loop, Rheodyne, Rohnert Park, CA, USA), a Supelcosil LC-18 column (250×4.6 mm; 5 µm; Supelco, Darmstadt, Germany) and a Jasco MD-1510 diode array detector (Cremella, Italy). The elution was performed with an isocratic mixture of acetonitrile and water (60:40), with a flow rate of 1.5 mL min^−1^. External calibration curves in six concentration levels were constructed from standard solutions. 

The gravimetric quantification of PM_2.5_ was performed on a RADWAG MYA 5/2Y/F (Radom, Poland) microbalance with an accuracy of 1 μg in a humidity (50 ± 5%) and temperature (20 ± 1 °C) controlled room. Filter weights were obtained from the average of six consecutive measurements with variations between them of less than 0.02%. The carbonaceous content (organic and elemental carbon, OC and EC) of PM_2.5_ samples was analysed by a thermal-optical transmission technique. At least, two replicate analyses were performed for each filter. In each analytical run, two 9 mm punches are first heated in a non-oxidising atmosphere of N_2_ in order to volatilise the carbonaceous organic compounds. After the first step of controlled heating, the remaining carbonaceous fraction is burnt in an oxygen-containing gas mixture. During anoxic heating, some OC is pyrolysed (PC), and quantified as EC in the second stage of heating. The minimise the bias in the OC/EC split, the blackening of the filter is continuously monitored by a laser beam and a photodetector, which allows reading the light transmittance. OC and EC are measured in the form of CO_2_ by an infrared non-dispersive analyser. 

For chemical and morphological characterisation, the filters with the lowest and highest concentration in each kitchen and the respective outdoor pairs were chosen. Two 5 mm diameter punches were cut from each of these filters. A Hitachi S-4100 scanning electron microscope (SEM) coupled to a Bruker Quantax 400 Energy Dispersive Spectrometer (EDS) (Bonsai Advanced, Madrid, Spain) was employed. 

Since several punches were removed from each filter for analysis of the carbonaceous material and for morphological characterisation, the remaining area did not contain enough mass for the quantification of PAHs. Thus, for each site, the leftover area of the various filters was combined and extracted together to obtain an “average” of the concentrations. Each set of filters was extracted three times with dichloromethane (DCM) in an ultrasonic bath (25 mL for 15 min, each extraction, with 5 min stops between them). After each extraction, the 3 DCM organic extracts of each composite sample were combined, filtered through pre-cleaned cotton and concentrated to a volume of 0.5 mL using a Turbo Vap® II evaporation system (Biotage, Charlotte, NC, USA). The concentrated samples were transferred into vials and dried under a gentle nitrogen stream. The extracts were analysed in a gas chromatographer-mass spectrometer (GC-MS) from Agilent (Santa Clara, CA, USA) with single quadrupole. The chromatographic system (GC model 7890B, MS model 5977A) was equipped with a CombiPAL autosampler (Agilent, Santa Clara, CA, USA) and a TRB-5MS (60 m × 0.25 mm × 0.25 μm) column (Teknokroma, Barcelona, Spain). The quantitative analysis was performed by single ion monitoring (SIM). Blank filters were analysed in the same way to obtain blank-corrected results. Data were acquired in the electron impact (EI) mode (70 eV). The oven temperature programme was as follows: 60 °C (1 min), 60 to 150 °C (10 °C min^−1^), 150 to 290 °C (5 °C min^−1^), 290 °C (30 min) and using helium as carrier gas at 1.2 mL min^−1^. The following mixture of deuterated internal standards (IS) was used to quantify PAHs: 1,4-dichlorobenzene-d4, naphtalene-d8, acenaphthene-d10, phenanthrene-d10, chrysene-d12, perylene-d12, fluorene-d10 and benzo[a]pyrene-d12 (Supelco). In the case of plasticisers, deuterated diethyl phthalate-3,4,5,6-d4 and bis (2-ethylhexyl) phthalate-3,4,5,6-d4 (Supelco) were used as IS. Calibrations were performed with authentic standards (Sigma-Aldrich, St. Louis, MO, USA) at eight different concentration levels.

### 2.2. Data Analysis

For the statistical treatment, SPSS (IBM Statistics Software V.25, Armonk, NY, USA) was used. The normality of the data was assessed by the Shapiro-Wilk test. The Mann-Witney non-parametric test was applied to obtain the statistically significant differences with a significance of 0.05 (Appendix A). Uncertainties of measurements were estimated as 5/6 times the method detection limit, which is a common procedure adopted in factor analysis. On average, the absolute uncertainties for PM_2.5_, OC and EC were 0.40, 0.14 and 0.13 µg m^−3^, which correspond to relative errors of 1.4–2.9%, 1.8–4.4% and 2.0–5.8%, respectively. For organic compounds, depending on the PAH or plasticiser, uncertainties were estimated to be in the range from 1.2 to 25 pg m^−3^, accounting for relative errors of 1.3–5.2%. In the case of volatile organic compounds, individual uncertainties were always < 0.1 µg m^−3^ with relative errors ranging from 0.28 to 6.6%.

### 2.3. Health Risk Assessment

To estimate the risk associated with inhalation of pollutants, the methodology proposed by the United States Protection Agency (USEPA), and extensively described in the literature (e.g.) [41], was followed. The assessment refers only to the period at home, since information on the time spent at work or other microenvironments or outdoors was not available. Given that several studies suggest that there are no significant differences between the levels in the various subcompartments of the residential dwelling (e.g.) [42,43], measurements in the kitchens were taken as representative of exposure at home. To account for the permanence in each household, time-adjusted concentrations (E_i_) were calculated using the following equation:(1)$$E_{i}=\underset{j}{\sum }C_{\mathrm{ij}}t_{j}\times \frac{\mathrm{EF}}{\mathrm{NY}}\times \frac{\mathrm{ED}}{\mathrm{AL}}$$
where E_i_ is the time-weighted daily personal exposure to compound i (μg m^−3^), C_ij_ is the measured concentration of compound i (μg m^−3^) in each household, t_j_ is the time fraction spent at home, EF is the exposure frequency (350 days/year considering that people spend 15 days on vacation away from home), NY is the number of days per year (365 days/year), ED is the exposure duration (30 years), and AL is the average lifetime (70 years). 

The inhalation unit risk (IUR) is the excess cancer risk resulting from continuous exposure to a unit increase of a compound via inhalation. IUR values listed in Table 2 are derived from previous studies by the USEPA for the general population with a default body weight of 70 kg and a default inhalation rate of 20 m^3^ day^−1^. The chronic inhalation cancer risk (CR) is the increased probability of developing cancer as a result of a specific exposure to a certain compound. CR is calculated using the following equation:CR_i_ = E_i_ × IUR_i_(2)

Cancer risks < 1 in a million are considered negligible, whereas values above 1.0 × 10^−4^ are classically considered of concern.

The inhalation non-cancer risk is estimated as follows:HQ_i_ = E_i_/R_f_C_i_(3)
where HQ_i_ is the hazard quotient of compound i, and R_f_C_i_ is the chronic reference concentration of compound i in μg m^−3^ (Table 2). The hazard index (HI) is the summation of non-cancer risks from multiple compounds. Values higher than 1 express a chance that non-carcinogenic effects may happen, whilst values below 1 indicate low or no risk of non-carcinogenic effects on humans.

The carcinogenic risk due to exposure to PAHs is based on benzo[a]pyrene equivalent concentrations (BaP_eq_). These are calculated multiplying the individual PAH concentrations by their toxic equivalent factor (TEF) [44]. The inhalation cancer unit risk of BaP is 1.11 × 10^−6^ (ng m^−3^) ^−1^. It is estimated from the cancer potency factor (CPF) using the following equation:IUR = CPF × 20 m^3^/(70 kg × 10^6^)(4)
where inhalation unit risk (IUR) represents the excess cancer risk associated with an exposure to a concentration of 1 µg m^−3^, CPF (equal to 3.9 (mg/kg-day) ^−1^ for BaP) indicates the excess cancer risk for an exposure to 1 mg of a compound per kg of body weight (70 kg), 20 m^3^ is the default inhalation rate per day, and 10^6^ is the conversion factor from mg to ng. The excess cancer risk for a receptor exposed to PAHs via the inhalation pathway can be estimated by equation 2, where E_i_ represents the time-weighted daily personal exposure to the sum of BaP_eq_ concentrations.

## 3. Results and Discussion 

### 3.1. Carbonyls and Volatile Organic Compounds 

Formaldehyde (HCHO) and acetaldehyde (CH_3_CHO) are highly reactive carbonyl compounds that are normally found in both indoor and outdoor environments. Formaldehyde is emitted by various building and insulating materials, some consumer products (e.g. disinfectants and cosmetics), carpets, fabrics and new furniture, principally if made of plywood [45,46]. In indoor environments, combustion processes, including tobacco smoking, also emit large amounts of these compounds. Acetaldehyde is also present in various consumer products such as deodorants, and in many foods and alcoholic drinks [45], which can represent emission sources in kitchens.

Indoor and outdoor formaldehyde concentrations were 7.61 ± 3.08 and 1.49 ± 0.67 µg m^−3^, respectively, whilst the corresponding acetaldehyde levels were 7.94 ± 4.63 and 0.41 ± 0.36 µg m^−3^ (Table 3). 

Formaldehyde levels in the kitchen of the detached house on the outskirts were found to be statistically different (*p* < 0.0146) from those in houses 1 (rural) and 2 (city centre apartment). The acetaldehyde levels of house 2 differed significantly from the values obtained in house 1 (*p* = 0.0224). Concentrations in kitchens were much higher than those observed outdoors. Statistically significant differences between the values of the indoor and outdoor environments were registered (*p* = 0.0001 for formaldehyde, *p* = 0.0004 for acetaldehyde, α = 0.05). These carbonyls are known to be irritants of the eyes and upper airways. Formaldehyde is a known human carcinogen [48]. In the present study, its concentrations never exceeded the protection limit of 100 µg m^−3^ imposed by the Portuguese legislation. Acetaldehyde was incorporated by the WHO in Group 2, which comprises pollutants of potential interest, but additional investigation would be needed before it is clear whether there is enough evidence to warrant their inclusion in the guidelines. Based on studies of short- and long-term exposure, countries such as Canada have set a maximum daily limit of 280 μg m^−3^. An extended review of formaldehyde concentrations worldwide in all types of indoor environments has been compiled by Salthammer et al. [49]. In a study carried out in dwelling in Bari, Italy, indoor formaldehyde and acetaldehyde concentrations were found to be significantly higher than outdoor concentrations. No significant relation was observed by the authors between the levels of aldehydes in the kitchens and the age or restoration of the building, the time windows or balcony doors were kept open or the time the burners were kept alight [45]. These two carbonyl compounds were also assessed in three principal rooms of 61 flats in Paris [42]. Statistically, the levels monitored in the kitchens did not differ from those registered in bedrooms and living rooms.

Many of the sources that contribute to carbonyls are also emitters of other VOCs. A few of these VOCs, such as benzene, are designated by multiple authorities as human carcinogens. Short- and long-term exposures can affect many organs and cause multiple symptoms [50]. In the present study, for most compounds, all the kitchens registered indoor-to-outdoor VOC concentration ratios higher than one, proving the strong contribution of endogenous emission sources. Statistically significant differences were found between indoor and outdoor levels of toluene (*p* = 0.0236), ethylbenzene (*p* = 0.0397), *m* + *p*-xylene (*p* = 0.0273), styrene (*p* = 0.005), *o-*xylene (*p* = 0.0500) and α-pinene (*p* = 0.0001). The dwelling with the most significant differences for a greater number of compounds compared to the others was the permanently occupied apartment, where an elderly woman who needs nursing care at home resides. The concentrations of VOCs in the four dwellings are within the wide range of values measured in homes of several other regions [41,43,51,52,53], although closer to the lower levels. The indoor concentrations of benzene, trichloroethylene, toluene, styrene and tetrachloroethylene were well below the thresholds laid down by the national regulation (Table 4). 

Indoors, benzene, toluene, ethylbenzene and xylenes (BTEX) were highly correlated with each other (*r*^2^ from 0.64 to 1.0), suggesting common emission sources. Outdoors, toluene, ethylbenzene and xylenes correlated well (*r*^2^ from 0.73 to 0.97), but the relationships involving benzene were weaker. However, the indoor concentrations of the various compounds did not correlate with the respective outdoor levels, indicating that the emitting sources in the kitchens are different from those observed outside. The only exceptions were tetrachlorethylene (*r*^2^ = 0.95) and styrene (*r*^2^ = 0.68). Tetrachlorethylene is mainly used for dry cleaning of fabrics, whereas styrene occurs naturally in small amounts in some plants and foods, such as peanuts, cinnamon, and coffee beans, although it is mostly used to make products such as food containers, rubber, plastic, carpet backing, insulation, fiberglass, pipes, and automobile parts. The highest indoor concentration was generally observed for α-pinene, followed by toluene and *m* + *p*-xylene. α-Pinene concentrations were 16 to 152 times higher in the kitchens than outside. This monoterpene is mainly synthetised by plants and commonly incorporated as fragrance in several consumer products (e.g. cleaning agents and air fresheners). It is emitted from numerous indoor items, including furniture of wooden origin [55]. Furthermore, cooking with condiments has been reported to be an important source of terpenes in indoor environments [56]. α-Pinene was also the most abundant and frequently detected VOC in UK and Polish homes [55,57]. On average, a toluene-to-benzene (T/B) ratio of 2 was obtained for the outdoor samples, which is a typical value for traffic emissions [58]. The indoor T/B ratios were three times higher than those observed in outdoor air. Likewise, the indoor samples were characterised by *m* + *p*-xylene-to-benzene and ethylbenzene-to-benzene ratios of 4.9 and 1.5, on average, while the corresponding outdoor values were 1.3 and 0.4. Toluene and benzene are common constituents of gasoline. However, toluene, together with ethylbenzene and xylenes, is used in solvents, while benzene is not. Solvents are the main component of cleaning agents, coatings, paints, adhesives, etc. Thus, evaporative emissions from coated surfaces and cleaning products, among other sources, may have contributed to enhanced emissions of toluene, xylenes and ethylbenzene in the kitchens. The *m* + *p*-xylene-to-ethylbenzene (*mpX/E*) ratio is frequently employed as an indicator of the age of air masses at a given site [58], since *m,p*-xylene disappear more rapidly than ethylbenzene through photochemistry. Thus, a higher *mpX/E* ratio suggests fresh local emissions, whereas lower ratios are related to more photochemical activity and associated emissions from some distance. In the present study, *mpX/E* ratios presented very little variability, averaging 3.3 and 3.1 for indoor and outdoor samples, respectively. These values are in agreement with ratios of fresh in situ emissions [58].

### 3.2. PM_2.5_ Concentrations and Carbonaceous Content

The mean indoor PM_2.5_ concentration ranged from 13.8 µg m^−3^, in the kitchen located in a rural area, to 30.2 µg m^−3^ in the city centre apartment with permanent occupancy (Figure 1). 

The kitchen of this apartment was the one where the concentrations most often exceeded the WHO guideline value. As observed for VOCs, the PM_2.5_ levels of this kitchen were found to be significantly different from those measured in any of the other dwellings (*p* < 0.0262), suggesting that concentrations increase with the occupancy rate. A mean outdoor level of 18.3 µg m^−3^ was obtained in the rural area, while very close mean values were recorded in the centre and outskirts of the city (27.6–29.5 µg m^−3^). Much higher PM_2.5_ concentrations have been monitored in household kitchens where biomass fuels are used for cooking (Table 5). 

In these kitchens, it has been observed that particulate matter levels vary according to the fuel type: cow dung cakes > rice husk > agricultural residues > firewood > gas. Li et al. [61] evaluated the household concentrations of PM_2.5_ among urban residents of Lanzhou, China, concluding that changing from coal to gas or electricity could result in a reduction of PM_2.5_ in the kitchens by 40–70%. The application of a statistical test to the databases of the present study indicated that the PM_2.5_ concentrations in kitchens equipped with gas ranges are not statistically different from those in kitchens with electrical appliances (*p* = 0.486, α = 0.05).

Total carbon accounted for about 30% of the PM_2.5_ mass in the kitchens of the rural area and city centre apartment (Figure 2). In the kitchens of houses with less central location, but near roads with intense traffic, the TC/PM_2.5_ values were higher (40–50%). The corresponding outdoor mass fractions ranged, in general, between 20 and 40%, the highest values being registered at the two locations more influenced by traffic. In the kitchens, OC represented 30–35% of PM_2.5_, while lower mass fractions of this carbonaceous constituent (18–23%) were obtained in the outdoor air. In general, the indoor OC/EC ratios were higher than the corresponding outdoor values. Regardless of location, ratios >2 were usually observed. The highest OC/EC ratios were measured at the beginning of the sampling campaign, when the region was hit by wildfires. Measurements carried out in a busy roadway tunnel in central Lisbon exhibited an OC/EC ratio in a narrow range from 0.3 to 0.4, reflecting the composition of fresh vehicular exhaust emissions. Much higher ratios are indicative of secondary OC formation, biomass burning emissions, and cooking fumes [62] (and references therein). Additional sources that contribute to the organic carbonaceous component of PM_2.5_ in indoor air include paper and clothing fibres, microscopic specks of plastics, contaminants brought on the soles of our shoes, bacteria, skin flakes, cosmetics, cleaning products, etc. [63]. On the other hand, VOCs in indoor air react and form lower volatility reaction products. These reaction products may condense on existing particles or nucleate, producing secondary organic aerosols (SOA), which grow with time into larger particles. Surface chemistry can also be a source of indoor SOA [64].

PM_2.5_ outdoor concentrations were weakly or moderately correlated with indoor concentrations for three of the households (*r*^2^ = 0.23–0.53), while an excellent relationship (r^2^ = 0.96) was found for the terraced house in a residential neighbourhood, near a main road with intense traffic. Similar relationships were observed for EC. The indoor and outdoor OC levels correlated well for all households (*r*^2^ = 0.76–0.91), indicating common sources or formation processes. The slopes of the correlations between indoor and outdoor concentrations represent the infiltration factors, i.e., the fraction of the outdoor PM_2.5_ carbonaceous component that penetrates indoors and remains suspended [65]. It was estimated that from 32% (rural house) to 74% (city centre apartment with permanent occupancy and with a window often open) of the indoor OC was infiltrated from the outside. 

### 3.3. PM_2.5_-Bound Polycyclic Aromatic Hydrocarbons and Plasticisers

Eight phthalate plasticisers and one non-phthalate plasticizer [bis(2-ethylhexyl) adipate] were quantified in PM_2.5_ (Table 6). This type of compounds can be found in large quantities in plastics, vinyl flooring, varnishes, coating agents, sealing compounds, industrial and natural rubber articles, and adhesives. The finishing of textiles also relies on the use of flexibilising substances to improve their feel and pliability. Plasticisers can leak out of the different products, thus escaping into the environment, making them ubiquitous. Total concentrations in the kitchens ranged from 44 to 171 ng m^−3^. These values were three to 12 times higher than those detected in the outdoor air, reflecting the widespread employment of plasticised indoor materials. Total concentrations in the kitchens were found to be significantly different from those outdoors (*p* = 0.0343). The most abundant compounds were diisobutyl phthalate and bis (2-ethylhexyl) phthalate, together with di-*n*-butyl phthalate in two of the kitchens. The latter two compounds have been listed as major plasticisers in household dust [63]. Health concerns related to phthalate ester exposures have focused primarily on cancer and reproductive effects [66]. Evidence for an association between phthalate exposure and diabetes risk and obesity was also found [67]. Moreover, it has emerged that exposure to phthalates aggravate pulmonary function and airway inflammation in asthmatic children [68].

PAHs are prevalent environmental pollutants generated primarily during the incomplete combustion of organic materials. Except in the permanently occupied housing, PAH concentrations were much higher in the outdoor air. It should be noted that in this house a greater number of meals are prepared, and gas is used for cooking. The total concentrations obtained in the kitchens were statistically different from those found outdoors (*p* = 0.0499).

The highest PAH levels were obtained in the house under the influence of wildfires. In the outdoor air of this dwelling, the retene concentration deserves to be highlighted. This alkylated phenanthrene has been described as the most abundant polyaromatic in particulate matter samples from several wildfire events [69]. An overwhelming proportion of retene has also been found in the organic extracts of PM_2.5_ from the combustion of vegetal charcoal in barbecue grills [70]. More recently, it has been detected in non-exhaust particles resulting from tyre wear [71]. Retene in tyre-related samples may originate from the natural waxes and resins added as softeners and extenders to rubbers. In addition to biomass burning, this traffic non-exhaust source may justify the detection of retene in the outdoor air, which in part penetrates inside the buildings. In all samples, high PAH molecular weights with ≥ 4 rings dominated over lighter compounds, indicating prevalence of pyrogenic with respect to petrogenic sources.

Concentration ratios between PAHs have been frequently used as diagnostic tools to infer their sources [69,70,72]. In the present study, regardless of the location, BaA/(BaA + Chry) ratios around 0.4 were obtained, revealing a mixed contribution from petrogenic sources, cooking fumes and biomass burning emissions. Also, irrespective of the sampling site, a rather constant IP/(IP + BghiP) ratio around 0.5 was observed. Values ≥ 0.5 have been linked to wildfires, coal combustion and residential wood burning, while emissions from petroleum combustion are characterised by much lower ratios. Values between 0.4 and 0.5 have been described as typical of cooking emissions [73]. In contrast, the BaP/BghiP ratio showed some variations. The indoor ratios were always lower than those observed outdoors. The highest ratios were obtained in the suburban detached house, which is close to a charcoal grilled chicken restaurant without fume removal or scrubbing system. BaP/BghiP ratios > 1.2 have been pointed out as typical of both wildfires and coal combustion, whilst values around 0.4–0.5 are characteristic of vehicle emissions [69,70,72]. 

### 3.4. PM_2.5_ Morphological Characteristics

SEM images are widely used in the study of atmospheric particle morphology, and can directly show the particle size, shape, aggregation characteristics, composition, and even sources. The individual particle details could contribute to establish pollution tracers emitted by specific sources in future studies. The filters used in this study are made of quartz fibres with different diameters in a tree-dimensional filtration substrate. A blank/clean quartz filter (Figure 3a) was analysed by SEM in order to compare its microstructure with that of the filters on which the particles were collected. The particulate material was mainly composed of soot aggregates, fly ash particles and mineral particles, which mainly derive from combustion and dust. 

Figure 3b shows the diverse types of materials found in the outdoor environment of house 1 (rural) between 22 and 24 October 2017. Black arrows (Figure 3c) indicate some of the PM_2.5_ present in different depths of the quartz fibre filter and the size of the circles on the bottom right corner represents the aerodynamic diameter cut-off for PM_1_ and PM_2.5_ in a sample collected in the kitchen of house 2 (city centre apartment). The kitchen samples from house 3 (Figure 3d) also reveal a mix composition of different materials, reflecting the contribution from diverse sources, e.g. cooking activities and particle resuspension.

In outdoor samples, carbonaceous particles represent a significant amount of the total particulates, being soot aggregates the dominant carbonaceous material (Figure 4a,b). These soot masses are formed by ultrafine aggregates of spherical particles with nanometric variable sizes (50–200 nm). Previous studies suggest that these types of particles are formed during combustion processes, e.g. biomass, coal, and diesel [74]. The EDS analysis revealed carbon, oxygen and sulphur peaks typical from combustion. Indoor particles from households 1 and 2 also show soot aggregates with lower sulphur content, which are likely associated with the use of burner gas hobs in these two kitchens. Additionally, silicate and iron plerospheres and cenospheres (Figure 4c,d) fly ashes, with diameter ≤ 2 µm, were also found in outdoor samples, with Fe-Si-Cu-Al-Ca variable composition. In the outdoor sample of house 1 collected from 16 to 18 October, soot materials and fly ashes were more abundant than in other filters from the same location, possibly due to the occurrence of wildfires in those days in the nearby forests.

In indoor samples of house 1, several particles related to the peri urban/rural environment were found. Brochosomes (Figure 5a) are spherical honeycomb like particles, composed of proteins and lipids < 1 µm in diameter with which the leafhoppers (family Cicadellidae) coat themselves [75]. Also, in all houses, kitchen salt (NaCl) particles with dimensions <2.5 µm were abundantly found (Figure 3c and Figure 5b).

In summary, PM_2.5_ was not only comprised of irregularly shaped agglomerate particles but also contained spherical, elongated, and flocculent particles. It is known that spherical particles and soot aggregates can enable the fine particles to easily adsorb toxic and harmful substances, such as heavy metals, volatile organic contaminants, and semivolatile organic pollutants. The observation of many particles in the ultrafine mode, including in the nanoscale size range, is relevant from the point of view of health. In addition to being able to penetrate deeply into the airways, these particles have a high adhesion surface to adsorb various chemical constituents, resulting in an enhanced complexity and toxicity.

### 3.5. Cancer and Non-Cancer Risks

Regardless of household, the hazard quotients associated with VOC inhalation were always below 1, indicating a low probability of non-cancer effects (Figure 6). The total hazard index ranged from 0.40 to 0.64. As observed in previous works [41,43], formaldehyde and acetaldehyde were the compounds that contributed most to the total risk, accounting for 25–63% and 32–70% of HI, respectively. The global excess lifetime cancer risk varied between 2.7 × 10^−5^ and 4.7 × 10^−5^. Thus, CR was lower than the USEPA guideline of 1.0 × 10^−4^, but not negligible (>1 × 10^−6^). The major contribution to CR came, once again, from formaldehyde (59–81%) and acetaldehyde (8–35%). The highest risks were obtained in households located near roads with more intense traffic.

The cancer risks associated with domestic exposure to PAHs through the inhalation pathway ranged from 1.1 × 10^−7^ to 3.5 × 10^−7^, which can be taken as negligible. Benzo[a]pyrene accounted for more than a half (51–61%) of the total cancer risk, followed by dibenzo[a,h]anthracene and indeno[1,2,3-cd]pyrene with shares of 10–19% and 7–13%, respectively.

## 4. Conclusions

Particulate matter (PM_2.5_), VOCs and carbonyls were monitored in the indoor and outdoor air of modern kitchens of four houses. Except for benzene and tetrachloroethylene, all pollutants presented indoor-to-outdoor ratios higher than 1, demonstrating the contribution of domestic emission sources. Concentrations were lower than the thresholds stipulated in the legislation or recommended by international organisations and well below the values reported for kitchens in developing countries where solid fuels are used for cooking. The levels of both PM_2.5_ and VOCs in the kitchen of the permanently occupied home were significantly different from those observed in the other dwellings. Thus, it seems that a higher frequency of activities associated with full-time occupancy is the most determining factor for air quality. Carbonaceous constituents represented about 30–50% of the PM_2.5_ mass, with the highest mass fractions recorded in houses closest to high traffic routes. It was estimated that from 32% to 74% of the indoor OC penetrates from outdoors. OC/EC ratios were higher indoors than outdoors, always surpassing 2, and peaked when the region was plagued by wildfires. In outdoor samples, pherospheres and cenospheres fly ashes composed of Fe-Si-Cu-Al-Ca were abundant, while ultrafine soot aggregates represented the dominant carbonaceous material. Soot aggregates with lower sulphur content were also found in kitchens with burner gas hobs. Salt and mineral particles from soil resuspension were observed in all kitchens. Brochosomes were only detected in the kitchen of the rural house. Although the particle levels were found to be statistically different in only one of the dwellings, the PM_2.5_ morphology indicated the presence of particles with distinct properties in kitchens with gas cooking appliances compared to those equipped with electric hobs. 

Irrespective of the type of house, a low probability of non-cancer effects due to inhalation of VOCs was estimated. The global excess lifetime cancer risk was lower than the USEPA guideline of 1 × 10^−4^ but was higher than 1 × 10^−6^, so it cannot be considered negligible. Formaldehyde and acetaldehyde were the compounds that contributed most to the total cancer and non-cancer risks in the indoor environments. The cancer risk associated with residential exposure to particle-bound PAHs via inhalation was found to be insignificant. However, it is necessary to bear in mind that the morphological analysis revealed the presence of numerous ultrafine particles, including nanometric variable sizes, with a complex composition that comprises metals known to cause oxidative stress and other health hazards. 

Although logistically difficult, future studies should consider the analysis of other gaseous pollutants and a more detailed chemical characterisation of size distributed particulate material for a larger number of samples in order to be able to apply source apportionment models. The fact that concentrations are generally low does not offer a complete guarantee of health protection. For this reason, it is advisable that, in the future, chemical analyses be accompanied by in vitro toxicity testing to assess which constituents can be related to health impairment.

## Figures and Tables

**Figure 1 ijerph-17-05256-f001:**
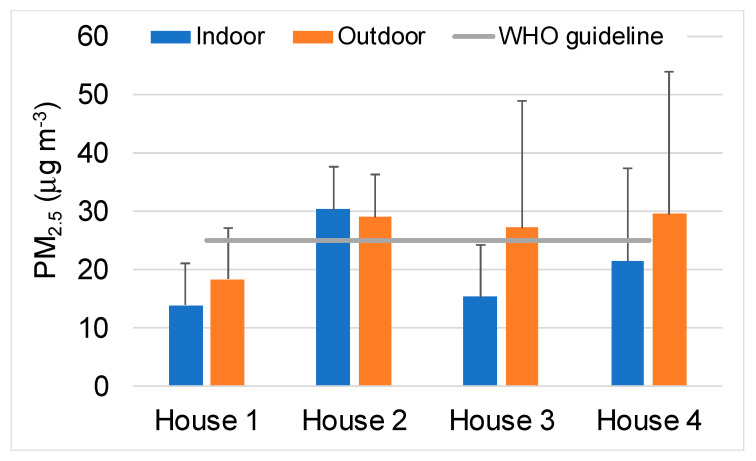
PM_2.5_ concentrations monitored in kitchens and outdoor air.

**Figure 2 ijerph-17-05256-f002:**
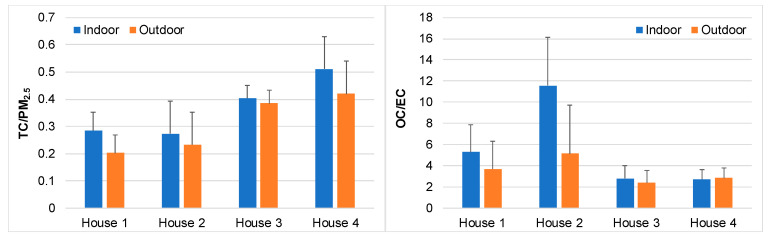
Ratios between total carbon (TC = EC + OC) and PM_2.5_ and between organic carbon and elemental carbon.

**Figure 3 ijerph-17-05256-f003:**
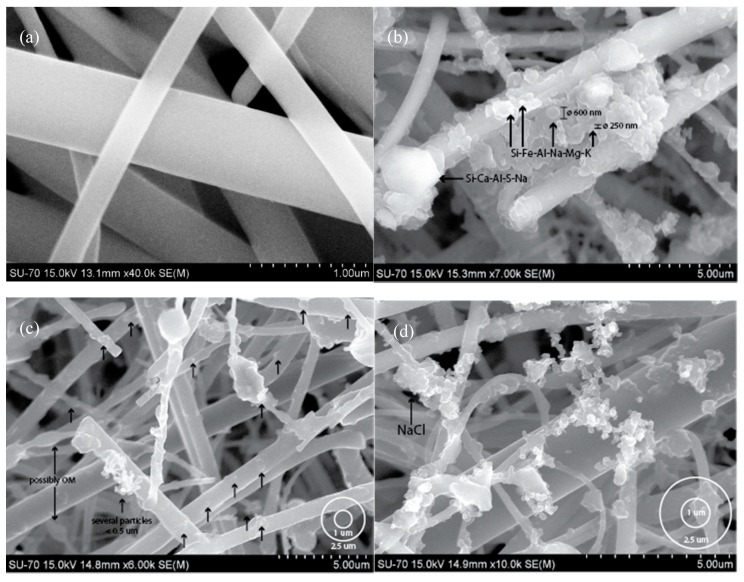
SEM imagens of (**a**) control blank quartz filter with 1.0 µm scale; (**b**–**d**) collected particles.

**Figure 4 ijerph-17-05256-f004:**
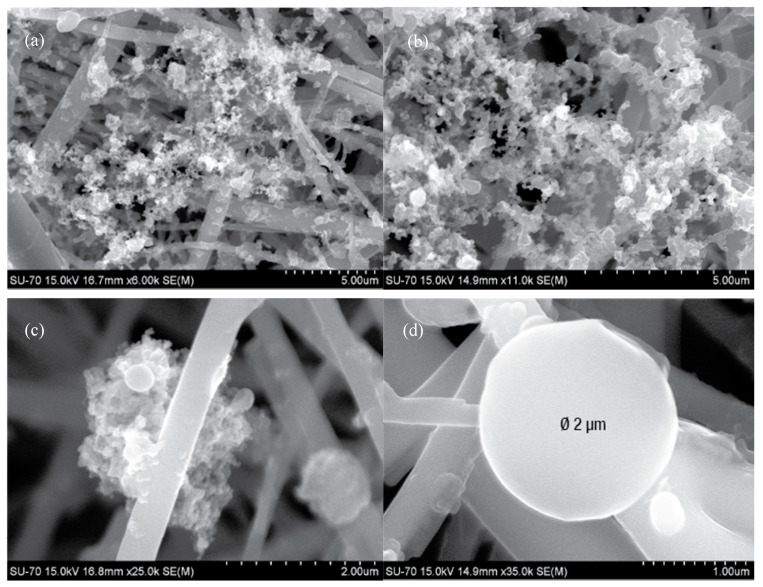
SEM images of soot aggregates with branching structures (**a**,**b**) and with cenosphere fly ash (**c**,**d**) in outdoor samples.

**Figure 5 ijerph-17-05256-f005:**
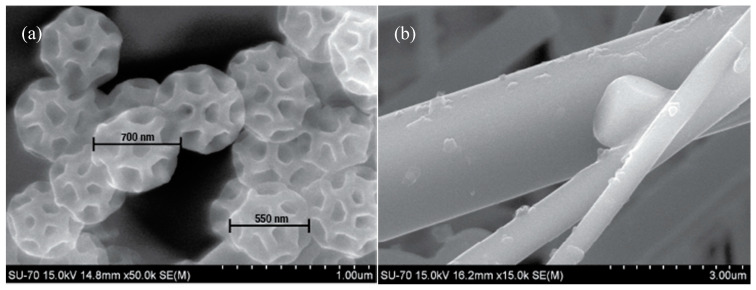
SEM imagens of (**a**) brochosomes, and (**b**) salt (NaCl) particles in indoor samples.

**Figure 6 ijerph-17-05256-f006:**
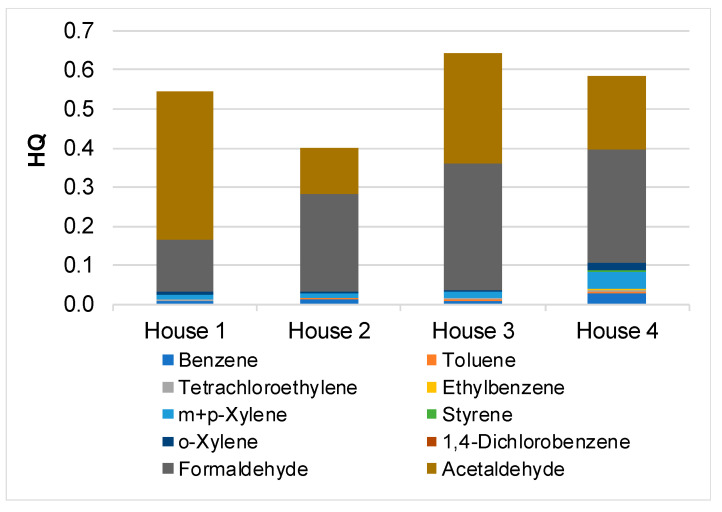
Hazard quotients (HQ) for individual VOCs for the four dwellings.

**Table 1 ijerph-17-05256-t001:** Characterisation of dwellings and sampling details.

Characteristics	House 1	House 2	House 3	House 4
Location	Ílhavo	Aveiro	Aveiro	Aveiro
Area	Periurban/rural	Urban	Suburban	Urban
Type of dwelling	Detached house with lawn and extensive vegetable garden	City centre apartment with permanent occupancy	Detached house on the outskirts, with small garden, near a main road with intense traffic	Terraced house in a residential neighbourhood, near a main road with intense traffic
Kitchen area (m^2^)	24.5	16.5	15.4	20.2
Number of permanent occupants	3	3	3	4
Number of daily occupancy hours	15	24	15	17
Smokers	No	No	No	No
Pets	1 dog, 1 cat	No	No	No
Source of energy for cooking	Gas	Gas	Electricity	Electricity
Ventilation	Natural	Natural	Natural	Natural
Range hood	Under cabinet	Under cabinet	Under cabinet	Under cabinet
PM_2.5_ sampling equipment	TCR Tecora	MiniVol	TCR Tecora	MiniVol
Number of samples	10 (indoor) 10 (outdoor)	7 (indoor) 7 (outdoor)	10 (indoor) 10 (outdoor)	7 (indoor) 7 (outdoor)
Sampling period	16/10/2017 to 05/11/2017	16/10/2017 to 06/11/2017	06 to 26/11/2017	07 to 28/11/2017

**Table 2 ijerph-17-05256-t002:** Toxicity parameters for VOCs provided by the Office of Environmental Health Hazard Assessment (OEHHA) and Integrated Risk Information System (IRIS) of USEPA (n.a.—not available).

Compound.	IUR (μg m^−3^) ^−1^	R_f_C (μg m^−3^)
Benzene	2.2 × 10^−6^	30
Toluene	n.a.	5 × 10^3^
Xylenes	n.a.	100
Ethylbenzene	2.5 × 10^−6^	1 × 10^3^
Styrene	1.63 × 10^−7^	900
Tetrachloroethylene	2.6 × 10^−7^	40
1,4-Dichlorobenzene	1.1 × 10^−5^	800
Formaldehyde	1.3 × 10^−5^	9
Acetaldehyde	2.7 × 10^−6^	9

**Table 3 ijerph-17-05256-t003:** Comparison of carbonyl concentrations (µg m^−3^) obtained in the present study with those reported for other places.

Location	Environment	Formaldehyde	Acetaldehyde	Reference
Aveiro region, Portugal	Kitchens	7.61 ± 3.08	7.94 ± 4.63	Present study
	Outdoor	1.49 ± 0.67	0.41 ± 0.36	
61 flats in Paris, France	Kitchens	21.7 ± 1.9	10.1 ± 1.8	[42]
Dwellings in Bari, Italy	Kitchens	16.0 ± 8.0	10.7 ± 8.8	[45]
	Outdoor	4.4 ± 1.7	3.4 ± 2.0	
59 homes in Prince Edward Island, Canada	Not provided	5.5–87.5 (median 29.6)	4.4–79.1 (median 29.6)	[46]
Shiraz, Iran	Outdoor-summer	15.1 ± 9.17	8.40 ± 4.29	[47]
	Outdoor-winter	8.57 ± 5.91	3.52 ± 1.69	

**Table 4 ijerph-17-05256-t004:** Minimum, maximum and mean values for VOC concentrations and I/O ratios, and legal limits.

VOC	Indoor Concentration Range and Mean µg m^−3^	Indoor-to-Outdoor Ratio	Threshold by the Portuguese Legislation [54] µg m^−3^
Benzene	0.78–3.3 (1.6)	0.39–1.3 (0.68)	5
Ethylbenzene	0.87–6.6 (2.4)	1.1–7.4 (2.9)	
Toluene	4.1–21.6 (9.4)	1.1–4.6 (2.3)	250
*m* + *p*-xylene	2.7–20.2 (7.6)	1.1–7.6 (3.1)	
*o*-Xylene	1.0–8.9 (3.1)	0.97–8.0 (3.0)	
Styrene	0.33–1.6 (1.0)	1.5–5.0 (3.5)	260
1,4-Dichlorobenzene	<0.10–1.2 (0.38)		25
Trichloroethylene	<0.10		
Tetrachloroethylene	0.30–2.9 (0.96)	0.56–1.2 (0.89)	250
α-Pinene	2.9–17.4 (9.5)	16.4–152 (71.1)	

**Table 5 ijerph-17-05256-t005:** PM_2.5_ concentrations measured in the present study and in kitchens of other countries using different cooking fuels or energy sources.

Location	PM_2.5_ (µg m^−3^)	Cooking Fuel or Energy Source	Reference
Aveiro region, Portugal	20.6 ± 10.9 17.8 ± 12.2	gas electricity	This study
Bhaktapur, Nepal	630 ± 924 759 ± 988 656 ± 924 169 ± 207 101 ± 130 80 ± 103	wood rice husk biomass mixture (wood + rice husk) kerosene LPG electricity	[59]
Rural households, India	910 447 432 78	dung cakes agricultural residues fuel mixture (wood + dung) LPG	[60]
Lanzhou, northwest China (heating season)	204 ± 50 114 ± 39 107 ± 43	coal gas electricity	[61]
Lanzhou, northwest China (non-heating season)	213 ± 89 65 ± 42 55 ± 35	coal gas electricity	[61]

**Table 6 ijerph-17-05256-t006:** Indoor and outdoor concentrations (ng m^−3^) of plasticisers and polyaromatic compounds and diagnostic ratios.

	House 1		House 2		House 3		House 4	
	Indoor	Outdoor	Indoor	Outdoor	Indoor	Outdoor	Indoor	Outdoor
*Plasticisers*								
Dimethyl phthalate	0.0993	0.00616	1.57	0.115	0.281	0.0105	0.916	0.119
Diethyl phthalate	1.49	0.0538	5.55	1.24	0.825		8.87	1.53
Diisobutyl phthalate	13.4	4.15	72.3	13.2	7.96	3.11	76.5	3.12
Di-*n*-butyl phthalate	1.44		30.9	3.92	2.80		56.8	
Benzyl butyl phthalate	0.547	0.394	2.05	0.160	0.302	0.135	0.580	0.132
Bis(2-ethylhexyl) adipate	7.93	2.70	6.34	0.847	3.20	2.18	6.11	0.985
Bis(2-ethylhexyl) phthalate	29.8	7.11	29.1	11.2	25.4	8.18	20.4	6.92
Di-*n*-octyl phthalate	2.95		1.16	0.113	1.91	0.254	0.747	0.294
Diisononyl phthalate	1.31	0.168	0.211	0.0376	1.17	0.0601	0.265	0.101
*Total*	59.0	14.6	149	30.8	43.8	13.9	171	13.2
*PAHs*								
Naphthalene	1.13	1.19	1.93	0.0192			1.81	1.22
Acenaphthylene		0.156	0.00896		0.0425	0.354	0.101	0.244
Acenaphthene				0.491				
Fluorene			0.0202			0.00557	0.0219	0.0387
Phenanthrene		0.0540	0.0411	0.000917	0.00745	0.185	0.0415	0.233
Anthracene		0.0132		0.00579	0.0159	0.0529	0.0473	0.0402
Fluoranthene		0.501	0.0484	0.0710	0.0963	0.666	0.183	0.602
Pyrene		0.656	0.0293	0.0534	0.0109	0.718	0.188	0.662
*p*-Terphenyl		0.0229		0.0110		0.0155		0.0168
Retene	0.106	5.26	0.568	1.31	0.117	0.621	0.136	0.581
Benzo[a]anthracene	0.0387	0.780	0.0705	0.0981	0.0604	1.13	0.194	0.731
Chrysene	0.0612	1.11	0.0950	0.162	0.0643	1.64	0.330	1.30
Benzo[b]fluoranthene	0.319	1.86	0.297	0.329	0.671	2.43	0.830	1.77
7,12-Dimethylbenz[a]anthracene	0.164	1.60	0.486	0.448	0.0910	0.298	0.0933	0.190
Benzo[k]fluoranthene	0.277	1.83	0.261	0.280	0.715	2.24	0.813	1.80
Benzo[e]pyrene	0.258	1.37	0.288	0.304	0.556	1.71	0.665	1.28
Benzo[a]pyrene	0.202	1.36	0.128	0.157	0.627	1.89	0.668	1.15
Perylene	0.108	0.719	0.0806	0.0920	0.301	0.875	0.314	0.528
Indeno[1,2,3-cd]pyrene	0.533	1.72	0.345	0.332	0.633	1.68	1.05	1.50
Dibenzo[a,h]anthracene	0.0749	0.245	0.0424	0.0389	0.0924	0.222	0.122	0.180
Benzo[g,h,i]perylene	0.460	1.47	0.359	0.345	0.547	1.52	1.05	1.41
*Total*	3.73	21.9	5.10	4.55	4.65	18.3	8.65	15.5
*Ratios between PAHs*								
BaA/(BaA + Chry)	0.39	0.41	0.43	0.38	0.48	0.41	0.37	0.36
IP/(IP + BghiP)	0.54	0.54	0.49	0.49	0.54	0.53	0.50	0.52
BaP/(BaP + BghiP)	0.36	0.48	0.45	0.47	0.50	0.53	0.39	0.48
BaP/BghiP	0.44	0.92	0.36	0.46	1.2	1.2	0.64	0.82

Empty cells mean below detection limit or of the same order of the blanks; BaA—Benzo[a]anthracene; Chry—Chrysene; IP—Indeno[1,2,3-cd]pyrene; BghiP—Benzo[g,h,i]perylene; BaP—Benzo[a]pyrene.

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
