# Peer review of "Fine Particulate Matter and Gaseous Compounds in Kitchens and Outdoor Air of Different Dwellings"

_ijerph, 2020, doi:10.3390/ijerph17145256_

Round 1

Reviewer 1 Report

It is a well prepared manuscript with lot of interest information.

1, Line 322, "Total carbon accounted for about 30% of the PM2.5 mass in the kitchens of the rural area and..."

It is nice to know TC/PM2.5, but in many times OC/PM2.5 is more important to evaluate the PM properties.

2, Line 335, "VOCs in indoor air may condense on existing particles..."

If authors look at data on the gas/particle partitioning coefficient of VOCs, VOCs will only have a very small percentage in the particulate matter phase. VOCs don’t condense on existing particles or nucleate to form SOA. However, the reaction products of VOCs will have lower volatility and condense on existing particles or nucleate to form SOA. The VOCs in indoor air do react and form lower volatility reaction products.

3, In Table 4

Some concentration numbers have one significant digit and some have 4 significant digit. Authors should be consistent to either following the rule of statistic analysis or to have the same number is significant digit after the point.

4, In the discussion, when comparing to the concentrations from other publications, to list many numbers from other publications is a distraction for readers. If authors can group those data into a table, it should be much better for readers and will also make the paper very neat.

Author Response

Item-by-item responses to the reviewers’ comments are provided in blue. Changes in the manuscript were yellow-shadowed. Manuscript in attach

It is a well-prepared manuscript with lot of interest information.

1, Line 322, "Total carbon accounted for about 30% of the PM2.5 mass in the kitchens of the rural area and..."

It is nice to know TC/PM2.5, but in many times OC/PM2.5 is more important to evaluate the PM properties.

A new sentence was included, providing information on the OC/PM2.5 mass fractions:

“In the kitchens, OC represented 30-35% of PM2.5, while lower mass fractions of this carbonaceous constituent (18-23%) were obtained in the outdoor air.”

2, Line 335, "VOCs in indoor air may condense on existing particles..."

If authors look at data on the gas/particle partitioning coefficient of VOCs, VOCs will only have a very small percentage in the particulate matter phase. VOCs don’t condense on existing particles or nucleate to form SOA. However, the reaction products of VOCs will have lower volatility and condense on existing particles or nucleate to form SOA. The VOCs in indoor air do react and form lower volatility reaction products.

The statement was rewritten, taking into account the reviewer’s comment:

“VOCs in indoor air react and form lower volatility reaction products. These reaction products may condense on existing particles…”

3, In Table 4

Some concentration numbers have one significant digit and some have 4 significant digit. Authors should be consistent to either following the rule of statistic analysis or to have the same number is significant digit after the point.

The same number of significant digits were provided for concentrations. Additional digits were included after the point aiming at always having 3 significant numbers.

4, In the discussion, when comparing to the concentrations from other publications, to list many numbers from other publications is a distraction for readers. If authors can group those data into a table, it should be much better for readers and will also make the paper very neat.

The discussion in section 3.2, where PM2.5 concentrations were compared with those of other studies, was significantly reduced and rephrased. Instead, a new table was created to list the values from the literature. Section 3.1 was also shortened and restructured to avoid an extended description of carbonyl concentrations from other studies. As suggested, data was grouped into a table.

Reviewer 2 Report

Dear authors,

Find attached my review.

Overall comments

The study presents a monitoring campaign carried out in four biomass-free kitchens and in the outdoor air. It was conducted in the region of Aveiro, Portugal, in October and 89 November 2017. The authors compared and evaluated VOC, carbonyl, and PM2.5 levels in the different dwellings. The problem presented and proposed solution in the paper is very interesting and actual. The paper itself is well written, although somewhat descriptive. The authors have conducted a thorough literature review, undertaken a rigorous piece of data collection. The comparison study makes comprehensive research on the existing literature of IAQ analysis. The authors also make a clear point on their scope, limiting the literature to studies made on VOC, carbonyl, and PM2.5 levels. The presented materials and method description is clear, but the results need improvements. In fact, the authors should provide more details about the statistical tests' comparison between different dwellings. This paper combines these multidisciplinary scientific fields. For this, I do think that the paper could be published with major revision. Unfortunately, I am a bit dubious about the lack of statistical analysis and results, which are vague and do not provide a clear resolution of the problem stated in the introduction. Particularly on gaining knowledge about the sources' contribution to the measured concentrations. The linear correlation used in this paper is a common pitfall in such a study. The problem of this paper is the statistical analysis of the data. This has to be resolved, otherwise, the paper does not seem to have a consistent problem-solution pair.

Specific Comments

Major comments

  1. I have several concerns, but the overarching one is that what is being proposed is not new. At least, the motivation was not highlighted enough. I have personally read dozens of papers about monitoring campaign results, and there are several reviews and cross-sectional studies. The readers need to answer to Why is the monitoring campaign worthwhile? Why isn't the problem already solved?
  2. The abstract (and/or Introduction) should include contributions and advantages.
  3. I think the motivations for this study need to be made clearer. In particular, the connection between (a) the source contribution, (b) the morphology of P M 2.5 , and the estimation of the health risk could be clearer.
  4. As far as I can tell, no comparisons were made between dwellings and definitely no statistical comparisons.
  5. The source contribution to the measured concentrations is somewhat descriptive. This is a generalization that cannot be made based on your study. It should, in general, compare the relative contribution, not the absolute one.
  6. Since the distribution of measurements in all experiments was not tested, the standard deviation values reported in Tables and text are of no statistical justification. Related, is important that the authors provide an uncertainty analysis of their results.
  7. There are many paragraphs that are unnecessary long. For example,
    1. lines 217 (Formaldehyde is... ) - 227(hydrocarbons),
    2. 253 (a few.. ) to 257 [50].

Minor comments

  1. Tables presentation: table row from splitting over two pages. It should be on the same page.

Author Response

Item-by-item responses to the reviewers’ comments are provided in blue. Changes in the manuscript were yellow-shadowed. Manuscript and supplementary material in attach.

The study presents a monitoring campaign carried out in four biomass-free kitchens and in the outdoor air. It was conducted in the region of Aveiro, Portugal, in October and 89 November 2017. The authors compared and evaluated VOC, carbonyl, and PM2.5 levels in the different dwellings. The problem presented and proposed solution in the paper is very interesting and actual. The paper itself is well written, although somewhat descriptive. The authors have conducted a thorough literature review, undertaken a rigorous piece of data collection. The comparison study makes comprehensive research on the existing literature of IAQ analysis. The authors also make a clear point on their scope, limiting the literature to studies made on VOC, carbonyl, and PM2.5 levels. The presented materials and method description is clear, but the results need improvements. In fact, the authors should provide more details about the statistical tests' comparison between different dwellings. This paper combines these multidisciplinary scientific fields. For this, I do think that the paper could be published with major revision. Unfortunately, I am a bit dubious about the lack of statistical analysis and results, which are vague and do not provide a clear resolution of the problem stated in the introduction. Particularly on gaining knowledge about the sources' contribution to the measured concentrations. The linear correlation used in this paper is a common pitfall in such a study. The problem of this paper is the statistical analysis of the data. This has to be resolved, otherwise, the paper does not seem to have a consistent problem-solution pair.

Specific Comments

Major comments

I have several concerns, but the overarching one is that what is being proposed is not new. At least, the motivation was not highlighted enough. I have personally read dozens of papers about monitoring campaign results, and there are several reviews and cross-sectional studies. The readers need to answer to Why is the monitoring campaign worthwhile? Why isn't the problem already solved?

The abstract (and/or Introduction) should include contributions and advantages.

I think the motivations for this study need to be made clearer. In particular, the connection between (a) the source contribution, (b) the morphology of PM2.5, and the estimation of the health risk could be clearer.

In fact, multiple studies have been carried out in kitchens where biomass is used as a cooking fuel. It is stated in the Introduction, that despite the pollutant levels in well-equipped modern kitchens are reportedly much lower, studies on this type of microenvironment are scarce and mostly focused on gaseous contaminants.

The abstract was not modified because of the very limited number of words imposed by the journal, but the last paragraph of the introduction has been changed to: “This study is based on a multi-pollutant monitoring campaign carried out in four biomass-free kitchens, for which studies are comparatively much scarcer, in order to answer the following questions: Are there significant differences in pollutant levels between modern kitchens equipped with gas stoves or electric hobs? Do the observed levels and compounds depend on housing factors or outdoor air? Are the risks resulting from inhalation of pollutants (VOCs and PM2.5-bond PAHs) routinely considered by international agencies of concern to health? Are these metrics sufficient to infer sources and effects or can the particle morphological analysis give us additional indications? The aim of this pilot study is not only to characterise air quality in a poorly studied microenvironment, such as kitchens, but also to draw lessons for conducting wider researches in the future.”

In order for the reader to better understand the relationship between the various subjects studied, several sections of the manuscript were expanded:

 Section 3.4.” SEM images are widely used in the study of atmospheric particle morphology, and can directly show the particle size, shape, aggregation characteristics, composition, and even sources The individual particle details could contribute to establish pollution tracers emitted by specific sources in future studies.”…” In summary, PM2.5 was not only comprised of irregularly shaped agglomerate particles but also contained spherical, elongated, and flocculent particles. It is known that spherical particles and soot aggregates can enable the fine particles to easily adsorb toxic and harmful substances, such as heavy metals, volatile organic contaminants, and semivolatile organic pollutants. The observation of many particles in the ultrafine mode, including in the nanoscale size range, is relevant from the point of view of health. In addition to being able to penetrate deeply into the airways, these particles have a high adhesion surface to adsorb various chemical constituents, resulting in an enhanced complexity and toxicity.”

Conclusions. “Although the particle levels were found to be statistically different in only one of the dwellings, the PM2.5 morphology indicated the presence of particles with distinct properties in kitchens with gas cooking appliances compared to those equipped with electric hobs.”…” The cancer risk associated with residential exposure to particle-bound PAHs via inhalation was found to be insignificant. However, it is necessary to bear in mind that the morphological analysis revealed the presence of numerous ultrafine particles, including nanometric variable sizes, with a complex composition that comprises metals known to cause oxidative stress and other health hazards.”

“Although logistically difficult, future studies should consider the analysis of other gaseous pollutants and a more detailed chemical characterisation of size distributed particulate material for a larger number of samples in order to be able to apply source apportionment models”.

As far as I can tell, no comparisons were made between dwellings and definitely no statistical comparisons.

In the revised version of the manuscript, the discussion was expanded, and the following statistical comparisons were made:

  • carbonyl levels in the different kitchens,
  • indoor versus outdoor carbonyl levels,
  • levels of individual VOCs in the different kitchens,
  • indoor versus outdoor concentrations of individual VOCs,
  • PM5 concentrations in the different kitchens,
  • PM5 concentrations in kitchens equipped with gas ranges versus in kitchens with electrical appliances,
  • indoor versus outdoor concentrations of plasticisers and PAHs.

Four new tables were created (Supplemental Material) with p-values for a 95% confidence interval.

The source contribution to the measured concentrations is somewhat descriptive. This is a generalization that cannot be made based on your study. It should, in general, compare the relative contribution, not the absolute one.

The most common receptor-oriented models, especially for airborne particulate matter, are based on statistical analysis of pollutant concentrations measured at a sampling site to infer the source-types and estimate their contributions to the measured concentrations. The three most widespread receptor models are principal component analysis (PCA), positive matrix factorisation (PMF) and chemical mass balance (CMB). CMB is the most commonly used method among those requiring a very detailed knowledge of sources and emission profiles. Although it has been extensively applied to outdoor air, and despite having the possibility of employment with a reduced number of samples, its use with indoor samples has been impractical since there are no extensive databases with the emission profiles of the various indoor sources. In the USA, SPECIATE is the EPA's repository of organic gas and particulate matter speciation profiles of pollution sources in outdoor air. A similar database of atmospheric particulate matter emission source profiles in Europe (SPECIEUROPE) is being developed by the Joint Research Centre. However, these databases contain source profiles of outdoor sources only, so they cannot be used as input to apply CMB to indoor measurements. Multivariate models that as opposed to CMB, do not require experimental source profiles as input (eigenvalue analysis and factor analysis) are widely popular. PCA and PMF require large monitoring datasets. Although the minimum number of samples is not consensual, as a rule of thumb it is accepted that the number of chemicals/elements multiplied by a factor of ten gives an appropriate sample size for the application of both PMF and PCA. Initially, our sampling campaign was designed to comply with this rule in order to enable the application of a source apportionment model. We placed 4 samplers in the first dwelling (2 indoors + 2 outdoors) to obtain parallel filters that would allow us to have enough material to perform a detailed chemical speciation of the particulate matter. However, we quickly realized that this would be unfeasible. Along with the discomfort shown by the daily invasion of their privacy, the owners have repeatedly complained about the noise caused by the sampling pumps. To proceed with sampling in the homes of the owners who had already given their consent, the compromise solution was to reduce the number of sampling days and use only two samplers (1 indoors + 1 outdoors). This hampered the implementation of some of the planned analyses. As it is mentioned in the experimental section of the manuscript, since several punches were removed from each filter for analysis of the carbonaceous material and for morphological characterisation, the remaining area did not contain enough mass for the quantification of PAHs or other constituents. Thus, for each site, the leftover area of the various filters was combined and extracted together to obtain an overall “average” of the concentrations. Contrary to what happens in outdoor environments, where high-volume sampling or parallel instruments can be used to obtain a high number of samples and enough particulate matter for chemical speciation, in indoor environments, especially those where people spend a significant part of their daily lives, several constraints arise, mainly related to privacy issues, space occupied by equipment and noise. For these reasons, in our sampling campaign it was not possible to obtain the desired number of samples to enable the application of source apportionment methodologies and to present the relative contributions. This was already reflected in the Conclusions section: “Although logistically difficult, future studies should consider the analysis of other gaseous pollutants and a more detailed chemical characterisation of the size distributed particulate material for a larger number of samples in order to be able to apply source apportionment models.”

Since the distribution of measurements in all experiments was not tested, the standard deviation values reported in Tables and text are of no statistical justification. Related, is important that the authors provide an uncertainty analysis of their results.

A new subsection (2.2. Data analysis) was created, as follows:

“For the statistical treatment, SPSS (IBM Statistics Software V.25) was used. The normality of the data was assessed by the Shapiro-Wilk test. The Mann-Witney non-parametric test was applied to obtain the statistically significant differences with a significance of 0.05. Uncertainties of measurements were estimated as 5/6 times the method detection limit, which is a common procedure adopted in factor analysis. On average, the absolute uncertainties for PM2.5, OC and EC were 0.40, 0.14 and 0.13 µg m-3, which correspond to relative errors of 1.4-2.9%, 1.8-4.4% and 2.0-5.8%, respectively. For organic compounds, depending on the PAH or plasticiser, uncertainties were estimated to be in the range from 1.2 to 25 pg m-3, accounting for relative errors of 1.3-5.2%. In the case of volatile organic compounds, individual uncertainties were always < 0.1 µg m-3 with relative errors ranging from 0.28 to 6.6%.

As stated before, throughout the manuscript, several statistical comparisons were made.

There are many paragraphs that are unnecessary long. For example, lines 217 (Formaldehyde is... ) – 227 (hydrocarbons), 253 (a few.. ) to 257 [50].

Following the suggestion of reviewer #1, the discussion in section 3.2, where PM2.5 concentrations were compared with those of other studies, was significantly reduced and rephrased. Instead, a new table was created to list the values from the literature. Section 3.1 was also shortened and restructured to avoid an extended description of carbonyl concentrations from other studies. As recommended by reviewer #1, data was grouped into a table. In addition, in section 3.1, the following part was deleted: “Formaldehyde and acetaldehyde largely contribute to the total VOC reactivity in the atmosphere, leading to the formation of new radicals, which, in turn, are involved in tropospheric ozone production. In urban areas, both compounds mainly originate from mobile sources. Biogenic sources comprise live and decaying plants, and seawater. Another source is secondary formation from the oxidation of hydrocarbons.” The previous sentence, mentioning the carbonyl sources, was simplified. The description of short- and long-term effects resulting from exposure to benzene was condensed in a single sentence: “Short- and long-term exposures can affect many organs and cause multiple symptoms”.

Minor comments

Tables presentation: table row from splitting over two pages. It should be on the same page.

Not much attention was paid to the layout of the tables because the original version of a manuscript changes after the review phase and the text is then subject to editorial typesetting. We can assure that the final version will be correctly formatted and that the tables will not be split between 2 pages.

Round 2

Reviewer 2 Report

Dear Authors

Thank you for this second round to review this manuscript. According to all responses of my comments,
the authors thoroughly revised the manuscript with a good critical summary of their results.
This revised manuscript has a high potential to be accepted in IJERPH.

In this version, the authors have conducted a thorough literature review, undertaken a rigorous piece of data collection, and have analyzed information accurately.

The manuscript is suitable for publication and only requires minor polishing;
thus, no further reviews are requested.

Best regards